# Variability in Catechin and Rutin Contents and Their Antioxidant Potential in Diverse Apple Genotypes

**DOI:** 10.3390/molecules24050943

**Published:** 2019-03-07

**Authors:** Wajida Shafi, Sheikh Mansoor, Sumira Jan, Desh Beer Singh, Mohsin Kazi, Mohammad Raish, Majed Alwadei, Javid Iqbal Mir, Parvaiz Ahmad

**Affiliations:** 1Indian Council of Agricultural and Research Central Institute of Temperate Horticulture, Old Airport Road, Rangreth, Srinagar 190007, J&K, India; wajida.shafi@gmail.com (W.S.); sumira.sam@gmail.com (S.J.); deshbsingh@yahoo.co.in (D.B.S.); 2Department of Biochemistry, Sher-e-Kashmir University of Agricultural Sciences and Technology, Jammu 180009, J&K, India; mansoorshafi21@gmail.com; 3Department of Pharmaceutics, College of Pharmacy, King Saud University, Riyadh 11451, Saudi Arabia; mkazi@ksu.edu.sa (M.K.); mraish@ksu.edu.sa (M.R.); ph.alwadei@gmail.com (M.A.); 4Botany and Microbiology Department, College of Science, King Saud University, P.O. Box. 2460, Riyadh 11451, Saudi Arabia; 5Department of Botany, S.P. College Srinagar, Srinagar 190001, Jammu and Kashmir, India

**Keywords:** apple, DPPH, FRAP, polyphenolics, catechins, rutin

## Abstract

Catechins and rutin are among the main metabolites found in apple fruit. Sixty apple genotypes, harvested in 2016 and 2017, were analyzed for their phenolic content and antioxidant activity. The HPLC analysis showed that the catechin concentration ranged from 109.98 to 5290.47 µg/g, and the rutin concentration ranged from 12.136 to 483.89 µg/g of apple fruit. The level of DPPH activity ranged from 9.04% to 77.57%, and almost half of the 15 genotypes showed below 30–40% DPPH activity. The apple genotypes ‘Lal Ambri’, ‘Green Sleeves’, and ‘*Mallus floribunda*’ showed the highest DPPH activity of between 70% and 80%, while ‘Schlomit’, ‘Luxtons Fortune’, ‘Mayaan’, ‘Ananas Retrine’, and ‘Chaubatia ambrose’ showed the lowest ferric reducing antioxidant power (FRAP) activity (0.02–0.09%). Statistical analysis showed a correlation between DPPH activity and catechin content (*r* = 0.7348) and rutin content (*r* = 0.1442). Regarding antioxidant activity, fractionated samples of apple genotypes revealed significant activity comparable to that of ascorbic acid. There was also a consistent trend for FRAP activity among all apple genotypes and a significant positive correlation between FRAP activity and rutin content (*r* = 0.244). Thus, this study reveals a significant variation in antioxidant potential among apple genotypes. This data could be useful for the development of new apple varieties with added phytochemicals by conventional and modern breeders.

## 1. Introduction

Apples are cultivated in temperate countries and are one of the most important fruits [1]. Worldwide, apples are consumed throughout the year because of their organoleptic qualities as well as due to technological advancements in the area of conservation [2]. Significant concentrations of phenolic compounds are present in apples and their products, and these play critical roles in maintaining human health due to their preventive effect against various diseases, such as cardiovascular diseases, neuropathies and diabetes [3]. The main phenolic acids found in apples are chlorogenic acid and *p*-coumaroylquinic acid, and the major flavonoids are epicatechins, catechins, procyanidins (B1 and B2), quercetin glycosides, anthocyanins, and phloridzin [4]. In recent years, there has been a rising inclination towards the use of bio-active compounds and in this context, extraction of such compounds from tissues rich in their content is desired [5,6]. Different plant materials have different extraction conditions, as they are affected by several parameters, such as the chemical nature of the sample, the type of solvent used, agitation, the time of extraction, the solute/solvent ratio and the presence of an optimum temperature [7,8]. Furthermore, validation of the extraction method for phenolic compounds is needed to avoid enzymatic oxidation during the process, as this leads to loss of phenol function and antioxidant potential [9]. Accordingly, in order to counteract oxidation, frozen or lyophilised samples are taken to prevent enzymatic oxidation [10].

Polyphenolic compounds are responsible for the aroma and organoleptic properties of apples. Phenolic acids in apples are subdivided into benzoic acids and hydroxycinnamic acids [11,12]. Flavonoids possess a nucleus comprising two phenolic rings and oxygenated heterocycle compounds, and they can be categorized into different types, e.g., anthocyanins, flavonols, flavanols (e.g., catechins), flavones, and chalcones [13]. Catechins and rutin are the predominant phytochemicals in apples; they not only confer color but also aroma to different genotypes. These compounds vary greatly in diverse apple genotypes depending on the place, season, light, and altitude [14,15]. The rationale of this study was to determine the variation in the concentrations of catechins and rutin in apple genotypes cultivated in the same location but harvested in two different years. Thus, the variation in catechin and rutin content and the antioxidant activity of apples were determined. The contribution of single phenolic compounds to the antioxidant capacity was estimated with special respect to standards. We intended to compare the antioxidant properties of catechins with those of other pure standards and synthetic antioxidants in all apple genotypes thriving in the same location under similar climatic and geographical conditions. The antioxidant activities were determined by commonly used methods of radical scavenging: DPPH (2,2-diphenyl-2-picrylhydrazyl) and ferric reducing antioxidant power (FRAP) assays.

## 2. Results

### 2.1. Phytochemical Determinations

#### Total Phenols, Total Flavanols, and Flavonoids

The total phenolic content of apples in this study ranged from 31.5 to 980.8 GAE/g, which is comparatively higher than the concentration in grape extract, a beverage known for its polyphenolic content. The flavanol content varied from 0.004 to 0.185 QEA (mg/g), while that of flavonoids ranged from 0.36 to 0.3584 QEA (mg/g). The maximum phenolic content, 980.8 GAE/g, was observed in the wild apple genotype *Mallus floribunda*, followed by 722.0 GAE/g in *Tydemans Early Worcestor*, and the minimum phenolic content of 31.5 mg L^−1^ was observed in Starking Delicious. The rest of the genotypes had moderate ranges. The maximum flavonoid content of 0.3884 QEA (mg/g) was observed in the wild apple genotype *Mallus floribunda* followed by *Ambri* (0.367 QEA (mg/g), and the minimum flavonoid content of 0.024 QEA (mg/g) was observed in Star Summer Gold, followed by 0.027 QEA (mg/g) in wealthy apple. The rest of the genotypes had moderate concentrations. Similarly, the maximum (0.351 QEA (mg/g)) and minimum (0.002 QEA (mg/g)) flavanol contents were observed in Orange Val and Red Fuji, respectively (Table 1).

### 2.2. Phytochemical Determinations

Quantification of Catechins and Rutin in Apple Genotypes by Reverse Phase-High Performance Liquid Chromatography (RP-HPLC)

Three types of polyphenol were detected in apple samples representing sixty apple genotypes (Table 2). Catechins and rutin were the predominant bioactive compounds in all apple genotypes. The maximum catechin content (5290.47 (µg/g)) was observed in the apple genotype *Malus floribunda* followed by Ambri (2028.94), and the minimum catechin content (238.57 (µg/g)) was observed in Neema Delicious, while the maximum (483.89 (µg/g)) and minimum (12.13 (µg/g)) rutin contents were found in Lal Ambri and June Eating, respectively.

### 2.3. Determination of the Antioxidant Potential of Apple Genotypes

With reference to the antioxidant potential of apple samples, the DPPH assay exhibited a minimum of 10.45% scavenging activity in Royal Delicious and a maximum of 77.57% in Michal, with Star Summer Gold presenting 39.10% scavenging activity on average (Table 1). Neema Delicious, Maharaji, and Ananas Retrine exhibited the lowest DPPH activity, while Benoni, Luxtons Fortune, Mayan and Chaubatia Ambrose exhibited the lowest FRAP activity. The antioxidant potential in terms of DPPH activity was found to be consistent among the apple genotypes, while the FRAP assay showed large variation among apple genotypes.

### 2.4. Correlation between Polyphenol Content and Antioxidant Assay

The antioxidant potential estimated by the DPPH assay, which involves an electron transfer mechanism from polyphenols, such as catechins and rutin, to DPPH, showed significant relations with rutin (*r* = 0.14424) and catechins (*r* = 0.7348). The FRAP assay, which measures the reducing potential of apple samples, also showed significant correlations with rutin (*r* = 0.244) and catechins (*r* = 0.9067). The most significant correlation was observed between FRAP and catechins (*r* = 0.9067) (Table 3).

### 2.5. Principal Component and Hierarchical Cluster Analysis of Bioactive Molecules and Antioxidant Assays

All observations recorded from the sixty apple genotypes were subjected to principal component analysis (PCA). The first three components explained 72.75% of the total variation (PC1 = 32.05, PC2 = 21.71%, and PC3 = 19.63%, respectively, Figure 1). The first principal component (PC1) was mainly contributed to by rutin and catechins, and PC3 was linked with the antioxidant assays DPPH (40.84%) and FRAP (67.33%). The PCA scatter plot revealed the distribution between 60 apple genotypes of diverse origins. The results obtained showed a comparatively discrete distribution of data points, thereby ascertaining that native apple genotypes, e.g., Ambri, Check Ambri, Lal Ambri, and Maharaji, together with wild ones such as *Malus baccata* and *Malus floribunda*, exhibit an extensive range of total antioxidant potential. The results for the native apple genotypes Ambri, Lal Ambri, and Red Delicious significantly deviated from those of other genotypes and also displayed the highest catechin and rutin contents and FRAP activity subsequent to the wild genotypes *Malus baccata* and *Malus floribunda*. The presence of the longest diagonal interception between catechins and rutin demonstrates that the higher disparity among apple genotypes and phenolic compounds varied significantly as a result of the apples’ diverse genetic backgrounds. Catechins and DPPH were closely associated, whereas rutin, flavanoids, flavanols, phenols, and FRAP were dispersed over the whole scatter plot, displaying a highly intricate association with regard to genotype.

Hierarchical Cluster analysis (HCA) analysis was carried out to evaluate the similarities between 60 apple genotypes, which were categorized into two main clusters (Figure 2): Cluster 1, which was characterized by relatively high FRAP activity, and Cluster 2, which was characterized by high levels of catechins and total phenols and had comparatively high DPPH activity. Cluster 2 was further divided into two clusters, with the wild genotype *Malusbaccata* exhibiting a close association with Red Delicious. Ambri and Lal Ambri shared the same sub-cluster under Cluster 1. Cluster 2 was divided into a sub-cluster bearing only the wild genotype *Malus Floribunda* and another sub-cluster comprising two clusters with genotypes such as Granny Smith, Chaubatia Ambrose, Check Ambri, and hybrid Shireen, which exhibited high FRAP activity.

## 3. Discussion

The association between the prospective health effects of apples and their flavor and aroma still remains very controversial. It would be exciting if the most flavorsome apple extract offered the most abundant bioactive metabolites and the highest antioxidant potential [16]. In the present research, in terms of the general quality of the extracts of diverse botanical origin, Cluster 2 presented the highest antioxidant potential and highest values for catechins and rutin, as estimated by both HPLC and UV-Visible spectrophotometry. Rutin is known to increase in some plants as the main strategy to protect against microclimatic variations [17]. The pronounced accumulation of rutin and catechins in our indigenous apple genotype, Ambri, puts it among the most desirable apple genotypes. Although the market demand for Ambri is still low, our data corroborate the possible health effects of extract. Even though the wild genotypes *Malus floribunda* and *Malusbaccata* accumulate catechins and rutin at maximum levels, due to their unpalatability as a result of high bitterness and astringency, they have low market acceptability [18]. However, these genotypes can function as parents to transfer desirable traits to diverse genotypes. Breeding programs with the aim of introgressing such fruits to commercial genotypes from wild species will play an important role in the development of new genetic material with higher contents of catechins and rutin. Additionally, the identification of molecular markers that are tightly linked to traits, such as high catechin and rutin content, could also pave the way for marker-assisted breeding programs in apples to transfer traits such as high catechin and rutin content. Hence, in this regard, it can be ascertained that Ambri containing a high content of polyphenols can be perceived as a nutritionally valuable extract. 

To better understand the data obtained and the functionality of the apple samples selected in this study, PCA and WCA were employed (Figure 1 and Figure 2). Using two-dimensional projections of apple genotypes procured by PCA, the clustering of extracts from the apples based on chemical composition and antioxidant power was likely; this projection was able to elucidate up to 75.93% of data variability. Apple extract exhibited the highest contents of total phenolic compounds, rutin, and catechins as well as the greatest antioxidant capacity in comparison with other extracts [19,20]. When WCA was applied to the whole data set and a Euclidean distance of 10.5 was considered, four different clusters were identified (Figure 2). Cluster 1 exhibited high levels of flavonoids and flavanols and low levels of phenolics as well as relatively high FRAP activity. Cluster 2 was characterized by high levels of catechins, total phenols and comparatively high DPPH activity. When variables were used for WCA, inferences about the associations among all responses could be made. The DPPH seemed to be highly associated with catechins, and rutin was strongly correlated with FRAP activity. 

Although we did not conduct a clinical study on apple extracts, research studies have determined the medicinal value of apples. The clinical significance of quercetin and catechins has been correlated with the antioxidant potential and radical scavenging activity of the apple samples. There is a significant disparity between the composition of phytochemicals such as catechins and quercetin in the different varieties of apple depending on the maturation and ripening stage of apple fruit [18]. Different research experiments have determined the relationship between the incidences of various diseases, such as cancer and coronary mortality, with daily intake of apples. There is a significantly inverse relationship between flavonoid intake and lung cancer development. The relationship between dietary catechins and epithelial cancer has been associated with 87% of the total catechin intake while apples contribute to 8.0% of catechin consumption [21]. 

Granato et al. [20] evaluated the chemical composition and antioxidant activity of Brazilian red wines by DPPH and FRAP assays and confirmed that flavonoids are the main phenolic class driving the antioxidant capacity. Further, HCA was applied to the variables and a quantitative measure of the degree of association between variables was performed using Pearson correlation coefficients; the results are presented in Table 3. Principal component analysis (PCA) is used to emphasize the variation and demonstrate strong patterns in a dataset [21]. PCA, using two antioxidant assays (DPPH and FRAP) and three antioxidants (catechins and rutin), showed that the first two components explained 78.31% of the total variation. Principal component (PC1) accounted for 83.04% of the total variation and the variables responsible for separation along the PC1 included DPPH (0.31), TP (0.25), rutin (0.43), and FRAP (0.40), and catechins (0.39). 

The hierarchical cluster analysis grouped 60 apple genotypes in clusters based on the antioxidant activity and anti-oxidative bioactive metabolites (Figure 2). The number of genotypes in the cluster varied from 2 (Cluster 4) to 24 (Cluster 2). The contribution of individual genotypes to the antioxidant grouping and the relationships between the clusters were assessed by plotting PC1 and PC2. PC1 and PC2 contributed average catechin values of 104.567 and 107.756, respectively, while PC3 was found to be rich in catechins, exhibiting a value of 111.859. On the contrary, PC2 was rich in rutin, displaying a value of 12.08, and PC3 was rich in catechins and antioxidants, 111.859 and 43.213, respectively. PC2 and PC4 were equal in terms of the average value of rutin accumulation in apple genotypes. Cluster 2, which was divided into sub-clusters, formed a separate group on the bi-plot and appeared to be distant from both Clusters 1 and 4. Like hierarchical clustering, accessions from cluster 4 grouped separately and formed a distant group on the PCA plot. Both PCA and cluster analysis were found to be equally effective for grouping the apple genotypes based on their antioxidant contents. The wild genotype *Malus floribunda* from Cluster 2 should be promoted for apple breeding programs and consumption because of its high antioxidant activities. Similar groupings of crop plants according to their antioxidant potential by means of cluster analysis and PCA have been done by other researchers worldwide [20,22].

## 4. Materials and Methods 

### 4.1. Materials

A total of 60 apple genotypes, originating from different geographical areas but cultivated under similar conditions, were obtained from the research farm of the Central Institute of Temperate Horticulture, Srinagar, India. The reagents used were Folin–Ciocalteau and DPPH (2,2-diphenyl-2-picrylhydrazyl). Methanol, acetone, and acetic acid were purchased from Hi media (Phillipsburg, NJ, USA). The aqueous solutions were prepared using ultra-pure water (Milli-Q, Millipore, São Paulo, SP, Brazil).

### 4.2. Methods

#### 4.2.1. Preparation of Extracts

##### Extraction of Phenolic Compounds (Catechin and Rutin)

The fragmentation of apples 10 fruits for each variety was carried out in a mortar and pestle, and fragments were immediately frozen with liquid nitrogen (1:2, *w*/*v*) in order to avoid oxidation of the phenolic compounds [23]. Homogenization of freeze-dried material without seeds was carried out by crushing in a mortar. An amount of 10 g of the powdered apple was transferred in Oakridge tubes and mixed with 10 mL of methanol or acetone in different concentrations, which was followed by incubation at −10 °C for 10 min. The mixture was centrifuged (8160× *g*, 20 min at 4 °C) (Sigma 3–30 K, Munich, Germany), concentrated by evaporation under vacuum (40 °C) in a rotary evaporator (IKA, HB-10, Germany), and freeze dried. The samples were reconstituted with 2 mL of 2.5% acetic acid and methanol (3:1, *v*/*v*) and filtered through a 0.22 µm (Nylon, Mumbai, India) syringe filter (Moxcare, Haryana, India) prior to analysis.

#### 4.2.2. HPLC Analysis

The HPLC analysis of samples was carried out in a Shimadzu HPLC (Kyoto, Japan) equipped with quaternary pumps, a degasser coupled to a photo-diode-array detector, and an injection valve with a 20 µL loop. An injection volume of 20 μL and a flow rate of 1.0 mL min^−1^ with 1 h of run time were used for the separation process. The analysis was carried out in triplicate for each sample. Chromatographic separations were performed on C18 (250 × 4.6 mm) with a 5 µm column using a solvent system in gradient mode followed by isocratic run, as represented in Table 4. The filtration of the mobile phase was done through a 0.45 μm membrane filter (Millipore, Bedford, MA, USA) and was subjected to 40 min ultrasonication. Instrument control, data acquisition, and data processing were done by using Class WP software (version 6.1) from Shimadzu (Columbia, SC, USA). Quantitative determinations were made by taking into account the respective peak areas of standards at a particular retention time versus the concentration and are expressed in mg/g of apple fruit.

#### 4.2.3. Determination of Total Polyphenolic Content (TPC)

The total phenol contents of extracts from different apple genotypes were determined by the modified Folin-Ciocalteau method [24]. Absorbance was then measured at 765 nm using a spectrophotometer. The results are expressed as mg of gallic acid equivalents μg/g FW.

#### 4.2.4. Determination of Total Flavonoid and Flavonol Contents

The total flavonoid content in apples (fruit) was determined using the method of Chang, et al. [25]. The absorbance was then measured at 415 nm using a spectrophotometer (Shimadzu, Columbia, SC, USA). Results are expressed in terms of the quercetin equivalent (mg/g). The same method was employed for flavonol determination, but the incubation period was 150 min instead of 40 min, and the absorbance was measured at 440 nm. The total flavonol content was also expressed in terms of the quercetin equivalent (mg/g).

### 4.3. Antioxidant Activity

#### 4.3.1. Ferric Reducing Antioxidant Potential (FRAP) Assay

The FRAP assay was done using the Benzie and Strain method with minor modifications [26]. The absorbance was then measured at 593 nm after 40 min. FeSO_4_ solution (0, 40, 80,160, 320, 640 µmol/L) was used for calibration of the standard curve. The results are expressed as μmol Fe^+2^ g^−1^ FW.

#### 4.3.2. DPPH (2,2-diphenyl-1-picrylhydrazyl) Scavenging Activity

DPPH free radical scavenging assay was measured using the procedure with slight modifications (Xu et al., 2012). Percentage inhibition was calculated by the formula(%IP) = [(A_t=0_ − A_t=15_)]/(A _t=0_) × 100(1)where A_t=15_ is the absorbance of the test sample after 15 min, and A_t=0_ is the absorbance of the control after 15 min.

Furthermore, the scavenging activity percentage (AA%) was determined.AA% = 100 − [(Abs_sample_ − Abs_control_)/Abs_blank_ × 100](2)where a mixture of methanol and DPPH in the ratio of 1:1 served as a blank, and a mixture of the standard (ascorbic acid) and DPPH in the ratio of 1:1 was used as the control. Here, the concentration of the test sample as well as that of the standard used was 15 µg/mL.

### 4.4. Statistical Analysis

In this study, the data was subjected to various statistical tests, such as cluster analysis, and correlations were determined to ascertain the superlative genotypes exhibiting high antioxidant potential estimated via DPPH and FRAP assays. All experiments were carried out in triplicate. The results are shown as mean values and standard error of the mean. The existence of significant differences among the results for total catechin and rutin contents was determined. The results obtained were subjected to one-way analysis of variance (ANOVA) and Duncan’s test. All statistical tests were done using SAS Enterprise Guide 4.2, SPSS 13 (SPSS Inc., New Orchard Road, NY, USA) and OP-STAT software (2.0, IBM, New Orchard Road, NY, USA) at a 5% significance level. Correlation analysis was carried out using Pearson’s test.

## 5. Conclusions

Overall, by using correlation analysis and multivariate statistical techniques (WCA and PCA), we verified that the phenolic compounds catechins and rutin are involved in the antioxidant activity of the commercial extract under study, although the contribution of other phytochemicals cannot be excluded. Apple samples from Ambri, Lal Ambri, and Red Delicious genotypes presented the highest antioxidant activity, as measured by antioxidant assays. On the other hand, apple extracts from Orange Val, Royal Delicious, and AnanasRetrine had the lowest DPPH and FRAP values. In this sense, the utilization of unsubstantiated statistical techniques, coupled to the ANOVA procedure, was demonstrated to be a suitable approach for evaluating the quality of commercial fruit extract based on various analytical measurements.

## Figures and Tables

**Figure 1 molecules-24-00943-f001:**
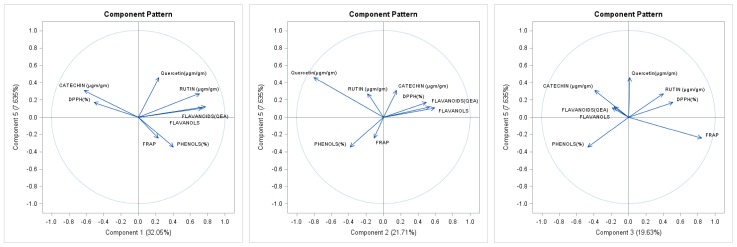
Principal component analysis showing the variability in phenolics and varied antioxidant assays among different apple varieties.

**Figure 2 molecules-24-00943-f002:**
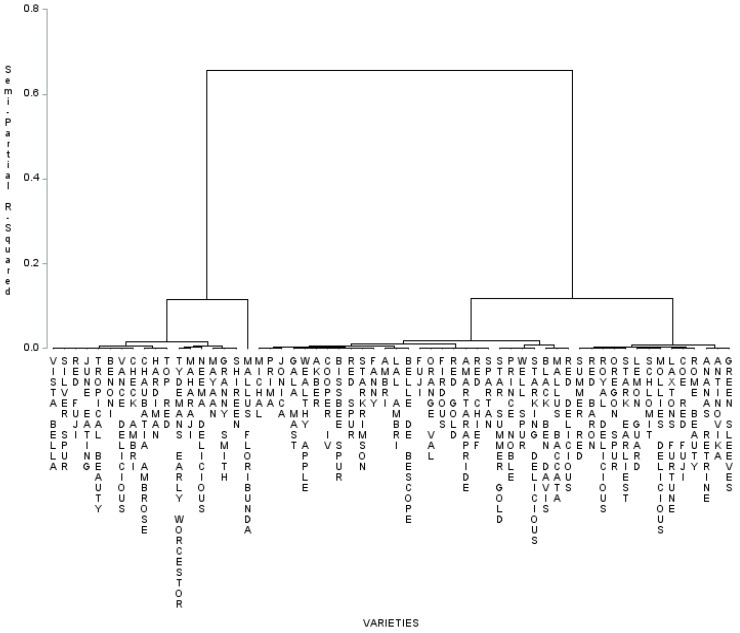
Schematic representation of 60 apple genotypes displaying the metabolically rich Cluster 2 (phenol rich/high DPPH), Cluster 5 (flavonoid rich/high FRAP), and Cluster 6 (quercetin rich) with their respective antioxidant potential values.

**Table 1 molecules-24-00943-t001:** Variability in rutin and catchin concentrations in different apple genotypes. The data is represented in mean ± SD (*n* = 10) and letter in the superscript symbolize the letters of significance with respect to each other using Tukey’ test.

S.NO.	VARIETIES	RUTIN (µg/g)	CATECHINS (µg/g)
1	VISTA BELLA	57.727 ^UT^ ± 5.87	1228.61 ^KLM^ ± 15.26
2	TYDEMANS EARLY WORCESTOR	87.023 ^R^ ± 5.34	665.10 ^XY^ ± 11.15
3	BENONI	28.70 ^ABC^ ± 2.34	1745.92 ^ED^ ± 18.79
4	MICHAL	140.17 ^J^± 9.87	1805.47 ^D^ ± 19.04
5	SUMMER RED	131.29 ^LJK^ ± 7.87	109.98 ^C^ ± 4.56
6	LEMON GUARD	46.86 ^VWX^ ± 4.32	1157.21 ^MNOP^ ± 11.26
7	LAXTONS FURTUNE	136.13 ^JK^ ± 7.65	832.52 ^TU^ ± 10.86
8	MAYAAN	20.31C ^D^ ± 1.23	1438.02 ^HI^ ±12.34
9	JUNE EATING	12.136 ^D^ ±2.32	851.43 ^ST^ ± 11.08
10	MOLLIES DELICIOUS	88.603 ^R^ ±7.32	1250.30 ^KLM^ ±16.04
11	PRIMA	53.251 ^UV^ ± 4.87	1498.37 ^GH^ ± 12.45
12	STAR SUMMER GOLD	63.08 ^T^ ±6.12	824.47 ^TU^ ± 9.32
13	BLACK BEN DAVIS	28.459 ^ABC^ ±2.45	1940.44 ^C^ ± 20.12
14	GALA MAST	65.598 ^ST^ ±6.78	1199.00 ^LMNO^ ± 12.04
15	AKBER	72.423 ^S^ ± 7.02	1309.31 ^JK^ ± 14.05
16	RED BARON	100.588 ^P^ ±8.12	373.43 ^A^ ± 10.45
17	FANNY	40.529 ^XY^ ± 3.45	1209.10 ^LMN^ ± 14.98
18	MALLUS BACCATA	212.087 ^E^ ± 9.12	1771.83 ^ED^ ± 18.98
19	FUJI	30.211 ^ABZ^ ± 3.12	357.43 ^A^ ± 11.08
20	VANCE DELICIOUS	124.454 ^LM^ ± 6.89	1409.72 ^I^ ± 12.01
21	COE RED FUJI	24.453 ^BC^ ± 2.13	982.03 ^Q^ ± 10.20
22	COOPER IV	59.31 ^UT^ ± 5.01	1253.16 ^KL^ ± 15.89
23	GRANNY SMITH	245.318 ^D^ ± 9.89	1608.65 ^F^ ± 17.78
24	AMBRI	332.405 ^B^ ± 10.98	2028.94 ^B^ ± 22.34
25	LAL AMBRI	483.888 ^A^ ± 11.23	1717.66 ^E^ ± 17.89
26	RED DELICIOUS	115.691 ^MN^ ± 7.08	1810.28 ^D^ ± 18.09
27	MALLUS FLORIBUNDA	85.961 ^R^ ± 7.23	5290.47 ^A^ ± 34.43
28	AMARTARAPRIDE	148.792 ^I^ ± 8.07	975.51 ^Q^ ± 12.03
29	ANANAS RETRINE	113.684 ^N^ ± 8.98	940.90 ^QR^ ± 11.87
30	ANTINOVIKA	36.763 ^AYZ^ ± 3.45	1180.84 ^LMNO^ ± 21.08
31	BELLE DE BESCOPE	101.316 ^OP^ ± 9.06	1239.24 ^KLM^ ± 15.78
32	BISSBEE SPUR	160.584 ^H^ ± 9.87	1117.38 ^OP^ ± 10.98
33	CHAUBATIA AMBROSE	124.388 ^LM^ ± 6.98	381.21 ^A^ ± 11.56
34	CHECK AMBRI	29.883 ^ABZ^ ± 3.01	1499.82 ^GH^ ± 12.01
35	FIRDOUS	200.225 ^F^ ± 8.98	366.60 ^A^ ± 9.89
36	GREEN SLEEVES	43.533 ^WXY^ ± 3.89	1085.48 ^P^ ± 11.05
37	HARDIMAN	119.226 ^MN^ ± 7.19	723.42 ^WXY^ ± 7.98
38	JONICA	50.889 ^UVW^ ± 4.56	1533.91 ^FG^ ± 15.32
39	MAHARAJI	240.65 ^D^ ± 9.06	863.66 ^RST^ ± 10.09
40	NEEMA DELICIOUS	86.995 ^R^ ± 8.12	238.57 ^B^ ± 10.04
41	ORANGE VAL	166.55 ^H^ ± 9.89	691.49 ^XY^ ± 6.78
42	OREGON SPUR	94.541 ^PQR^ ± 8.98	675.30 ^XY^ ± 5.98
43	PRINCE NOBLE	58.415 ^UT^ ± 5.10	416.41 ^A^ ± 12.09
44	RED CHIEF	112.096 ^N^ ± 8.08	640.32 ^Y^± 5.23
45	RED FUJI	330.08 ^B^ ± 10.09	1169.94 ^LMNOP^ ± 20.98
46	RED GOLD	47.608 ^WXV^ ± 4.56	328.76 ^A^ ± 10.09
47	RED SPUR	65.101 ^ST^ ± 6.28	1088.67 ^P^ ± 10.98
48	ROME BEAUTY	96.025 ^PQR^ ± 9.03	730.73 ^VWX^ ± 8.12
49	ROYAL DELICIOUS	87.085 ^R^ ± 7.79	547.12 ^Z^ ± 10.98
50	SCHLOMIT	98.579 ^PQ^ ± 8.76	1252.15 ^KL^ ± 17.86
51	SHIREEN	87.69 ^R^ ± 6.98	1371.73 ^IJ^ ± 18.78
52	SILVER SPUR	129.172 ^LK^ ± 7.46	1000.94 ^Q^ ± 18.78
53	SPARTAN	90.711 ^QR^ ± 8.90	782.16 ^TUVW^ ± 8.09
54	STARKRIMSON	38.951 ^XYZ^ ± 3.02	921.98 ^QRS^ ± 10.57
55	STARK EARLIEST	302.247 ^C^ ± 8.98	967.42 ^Q^ ± 10.45
56	STARKING DELICIOUS	109.75 ^NO^ ± 9.05	745.32 ^UVWX^ ± 7.16
57	TOP RED	167.432 ^H^ ± 9.14	361.08 ^A^ ± 9.08
58	TROPICAL BEAUTY	58.411 ^TU^ ± 4.78	1130.16 ^NOP^ ± 15.98
59	WEALTHY APPLE	179.283 ^G^ ± 6.98	1210.38 ^LMN^ ± 14.23
60	WELL SPUR	309.153 ^C^ ± 9.12	816.61 ^TUV^ ± 9.76

**Table 2 molecules-24-00943-t002:** Variability in antioxidant efficacy of diverse apple genotypes. The data is represented in mean ± SD (*n* = 10) and letter in the superscript symbolize the letters of significance with respect to each other using Tukey’ test.

S.No.	VARIETIES	DPPH (%)	FRAP (μmol Fe^+2^ g^−1^ FW)	FLAVONOIDS (QEA)	FLAVANOLS QEA (mg/g)	PHENOLS GAE
1	VISTA BELLA	30.47 ± 7.23	1.48 ± 0.17	0.178 ± 0.09	0.185 ± 0.06	545 ± 14.56
2	TYDEMANS EARLY WORCESTOR	29.14 ± 3.28	0.504 ± 0.16	0.163 ± 0.04	0.122 ± 0.03	722 ± 23.89
3	BENONI	19.08 ± 1.86	0.062 ± 0.006	0.147 ± 0.07	0.141 ± 0.06	523 ± 14.68
4	MICHAL	36.12 ± 8.34	1.75 ± 0.15	0.063 ± 0.005	0.021 ± 0.003	174.8 ± 9.89
5	SUMMER RED	50.2 ± 9.32	1.005 ± 0.90	0.201 ± 0.10	0.114 ± 0.04	395.1 ± 10.98
6	LEMON GUARD	24.35 ± 3.78	0.039 ± 0.002	0.078 ± 0.0021	0.064 ± 0.01	356.1 ± 9.87
7	LAXTONS FURTUNE	34.46 ± 8.20	0.03 ± 0.002	0.0627 ± 0.0018	0.032 ± 0.006	272.7 ± 7.67
8	MAYAAN	29.54 ± 3.19	0.049 ± 0.005	0.105 ± 0.09	0.108 ± 0.02	618.9 ± 20.90
9	JUNE EATING	36.63 ± 3.56	0.297 ± 0.080	0.112 ± 0.07	0.13 ± 0.01	583.9 ± 15.10
10	MOLLIES DELICIOUS	47.87 ± 8.10	0.437 ± 0.097	0.045 ± 0.009	0.012 ± 0.04	311.2 ± 9.12
11	PRIMA	38.83 ± 3.78	1.348 ± 0.13	0.057 ± 0.007	0.039 ± 0.007	160.8 ± 7.89
12	STAR SUMMER GOLD	39.1 ± 9.01	0.485 ± 0.078	0.024 ± 0.0014	0.006 ± 0.0005	85.7 ± 5.16
13	BLACK BEN DAVIS	49.94 ± 8.98	0.398 ± 0.056	0.03 ± 0.0011	0.009 ± 0.0011	106.6 ± 4.56
14	GALA MAST	40.85 ± 8.08	0.311 ± 0.049	0.164 ± 0.0012	0.008 ± 0.00013	201 ± 8.08
15	AKBER	32.75 ± 2.98	1.41 ± 0.12	0.1364 ± 0.005	0.151 ± 0.00012	150.3 ± 7.67
16	RED BARON	17.72 ± 1.17	0.494 ± 0.054	0.087 ± 0.006	0.054 ± 0.0065	335.7 ± 9.09
17	FANNY	66.21 ± 8.76	1.082 ± 0.70	0.1225 ± 0.0010	0.075 ± 0.0045	115.4 ± 5.46
18	MALLUS BACCATA	37.05 ± 3.15	1.551 ± 0.18	0.3584 ± 0.0054	0.13 ± 0.0052	108.4 ± 6.12
19	FUJI	51.06 ± 6.45	1.563 ± 0.16	0.108 ± 0.0045	0.092 ± 0.0034	174.8 ± 8.10
20	VANCE DELICIOUS	44.66 ± 7.67	2.132 ± 0.18	0.0836 ± 0.0054	0.068 ± 0.0059	562.9 ± 14.34
21	COE RED FUJI	9.04 ± 1.23	0.462 ± 0.013	0.147 ± 0.009	0.057 ± 0.0008	281.5 ± 6.78
22	COOPER IV	35.69 ± 2.17	2.995 ± 0.034	0.196 ± 0.007	0.078 ± 0.0004	169.6 ± 5.78
23	GRANNY SMITH	18.34 ± 1.09	0.51 ± 0.010	0.0455 ± 0.0078	0.021 ± 0.00015	648.6 ± 19.87
24	AMBRI	75.73 ± 9.04	2.241 ± 0.134	0.360 ± 0.030	0.114 ± 0.0005	244.8 ± 5.76
25	LAL AMBRI	77.57 ± 9.87	2.639 ± 0.167	0.192 ± 0.0076	0.171 ± 0.0009	232.5 ± 5.10
26	RED DELICIOUS	14.69 ± 1.02	0.498 ± 0.010	0.0195 ± 0.006	0.005 ± 0.00012	75.2 ± 8.98
27	MALLUS FLORIBUNDA	70.32 ± 6.78	1.377 ± 0.13	0.387 ± 0.045	0.144 ± 0.0020	980.8 ± 24.56
28	AMARTARAPRIDE	38.65 ± 3.10	2.313 ± 0.178	0.201 ± 0.098	0.131 ± 0.007	212.6 ± 7.65
29	ANANAS RETRINE	10.52 ± 2.05	0.081 ± 0.009	0.159 ± 0.005	0.162 ± 0.0065	455.2 ± 13.20
30	ANTINOVIKA	40.78 ± 3.19	1.434 ± 0.19	0.049 ± 0.0012	0.018 ± 0.0015	383.6 ± 8.79
31	BELLE DE BESCOPE	49.43 ± 4.14	1.185 ± 0.16	0.1505 ± 0.004	0.121 ± 0.006	248.3 ± 5.87
32	BISSBEE SPUR	63.83 ± 6.76	2.433 ± 0.198	0.1188 ± 0.007	0.096 ± 0.0006	166.1 ± 4.98
33	CHAUBATIA AMBROSE	35.92 ± 2.98	0.092 ± 0.187	0.1406 ± 0.006	0.127 ± 0.0004	493.7 ± 14.1
34	CHECK AMBRI	49.29 ± 4.09	1.617 ± 0.20	0.1938 ± 0.009	0.103 ± 0.0006	577.6 ± 15.78
35	FIRDOUS	46.51 ± 7.89	2.186 ± 0.12	0.1792 ± 0.007	0.138 ± 0.0005	209.8 ± 6.78
36	GREEN SLEEVES	40.27 ± 3.10	0.793 ± 0.070	0.0899 ± 0.006	0.027 ± 0.0008	418.5 ± 10.89
37	HARDIMAN	30.83 ± 2.56	2.232 ± 1.23	0.078 ± 0.0009	0.036 ± 0.0007	541.6 ± 14.23
38	JONICA	13.07 ± 3.04	2.331 ± 1.40	0.0875 ± 0.0008	0.048 ± 0.0008	199.3 ± 6.89
39	MAHARAJI	19.5 ± 3.98	1.142 ± 0.25	0.135 ± 0.009	0.06 ± 0.0009	653.8 ± 20.78
40	NEEMA DELICIOUS	14.16 ± 2.98	0.446 ± 0.065	0.07 ± 0.006	0.045 ± 0.0007	618.9 ± 18.89
41	ORANGE VAL	30.63 ± 2.98	0.102 ± 0.004	0.178 ± 0.0069	0.351 ± 0.054	171.7 ± 5.23
42	OREGON SPUR	47.27 ± 8.05	3.750 ± 0.198	0.213 ± 0.099	0.129 ± 0.006	342.7 ± 6.78
43	PRINCE NOBLE	28.68 ± 3.98	0.535 ± 0.078	0.036 ± 0.0078	0.021 ± 0.0008	103.1 ± 7.12
44	RED CHIEF	40.02 ± 2.98	2.022 ± 0.098	0.12 ± 0.007	0.087 ± 0.0007	218.5 ± 4.98
45	RED FUJI	24.63 ± 3.43	1.096 ± 0.007	0.099 ± 0.0006	0.002 ± 0.0001	540.6 ± 13.98
46	RED GOLD	38.64 ± 2.99	1.163 ± 0.19	0.0924 ± 0.0006	0.092 ± 0.0003	225.9 ± 8.10
47	RED SPUR	38.07 ± 3.19	1.396 ± 0.17	0.117 ± 0.0004	0.072 ± 0.0002	141.6 ± 3.52
48	ROME BEAUTY	30.02 ± 1.98	1.384 ± 0.14	0.06 ± 0.0004	0.024 ± 0.0001	290.2 ± 5.23
49	ROYAL DELICIOUS	10.45 ± 2.34	1.610 ± 0.20	0.101 ± 0.0005	0.061 ± 0.0005	342.5 ± 6.89
50	SCHLOMIT	14.06 ± 3.78	0.028 ± 0.056	0.084 ± 0.0005	0.315 ± 0.0064	356.8 ± 7.23
51	SHIREEN	35.27 ± 2.54	0.511 ± 0.070	0.1845 ± 0.0009	0.126 ± 0.006	664.0 ± 21.09
52	SILVER SPUR	22.22 ± 3.19	0.469 ± 0.067	0.0952 ± 0.00067	0.063 ± 0.00056	536.7 ± 12.87
53	SPARTAN	37.98 ± 2.98	0.318 ± 0.045	0.174 ± 0.0006	0.122 ± 0.004	218.5 ± 7.23
54	STARKRIMSON	45.95 ± 7.67	0.491 ± 0.07	0.054 ± 0.0004	0.021 ± 0.0015	162.6 ± 4.10
55	STARK EARLIEST	36.56 ± 2.20	2.568 ± 1.87	0.153 ± 0.0007	0.105 ± 0.002	347.9 ± 6.10
56	STARKING DELICIOUS	26.76 ± 4.12	0.931 ± 0.078	0.072 ± 0.0008	0.051 ± 0.0001	31.5 ± 3.04
57	TOP RED	22.78 ± 3.10	2.427 ± 1.87	0.091 ± 0.00067	0.141 ± 0.0006	533.9 ± 14.56
58	TROPICAL BEAUTY	50.06 ± 9.67	2.334 ± 1.23	0.063 ± 0.00023	0.039 ± 0.00013	564.3 ± 16.10
59	WEALTHY APPLE	48.51 ± 8.17	0.331 ± 0.043	0.027 ± 0.0067	0.009 ± 0.0001	181.8 ± 4.13
60	WELL SPUR	38.40 ± 3.19	2.010 ± 0.005	0.165 ± 0.0014	0.096 ± 0.00013	129.4 ± 1.43
CD		0.911	0.053	0.005	0.004	11.915
SE(d)		0.459	0.027	0.002	0.002	6.009
SE(m)		0.325	0.019	0.002	0.001	4.249
CV		1.550	2.752	2.245	2.881	2.162

**Table 3 molecules-24-00943-t003:** Correlation matrix between total phenolics, diphenyl-2-picrylhydrazyl (DPPH), and the ferric reducing antioxidant power (FRAP) with catechin and rutin in diverse apple genotypes.

	DPPH	PHENOLS	FLAVANOLS	FLAVONOIDS	FRAP	RUTIN	CATECHIN
DPPH		0.3789	0.0782	0.3528	0.3238	0.14424	0.7348
PHENOL			0.3023	0.2616	0.4241	0.8851	0.8614
FLAVANOLS				0.79049	0.3082	0.7655	0.6642
FLAVONOIDS					0.1899	0.802	0.6534
FRAP						0.24479	0.9067

**Table 4 molecules-24-00943-t004:** The table represents mobile phase and their gradient mode.

Compound	Mobile Phase	Gradient	λ _max_
Rutin Catechins	Solvent A—2.5% acetic acid Solvent B—aceto-nitrile	3–9% B (0–5 min)	280 320 350
		9–16% B (5–15 min)	
		16–36.4% B (15–33 min)	
		100% B (5 min)	

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
