# Peer review of "Variability in Catechin and Rutin Contents and Their Antioxidant Potential in Diverse Apple Genotypes"

_molecules, 2019, doi:10.3390/molecules24050943_

Round 1
Reviewer 1 Report
The present manuscript analyses the different content in catechin and rutin and the antioxidant activity of different apples. It would be desirable and more accurate to analyse the antioxidant activity also by ORAC and ABTS, and a deeper discussion of the results would be appreciated by the reader. In addition, in the text it is referred to certain supplementary material, but this referee has not been able to found it. Other points that should be checked:
Please, provide the complete name before using an abbreviation.
In table 1, what do the upper-index letters mean?
In certain occasion the authors refers to apple juice (also in the conclusion), but was not the extract of the fruit analysed? Please, explain.
Which was the purity of the standards (page 10, line 244)?
Author Response
Reviewer 1
Open Review
(x) I would not like to sign my review report
( ) I would like to sign my review report
English language and style
(x) Extensive editing of English language and style required
( ) Moderate English changes required
( ) English language and style are fine/minor spell check required
( ) I don't feel qualified to judge about the English language and style
Yes | Can be improved | Must be improved | Not applicable | |
Does the introduction provide sufficient background and include all relevant references? | (x) | ( ) | ( ) | ( ) |
Is the research design appropriate? | (x) | ( ) | ( ) | ( ) |
Are the methods adequately described? | (x) | ( ) | ( ) | ( ) |
Are the results clearly presented? | ( ) | (x) | ( ) | ( ) |
Are the conclusions supported by the results? | ( ) | (x) | ( ) | ( ) |
Comments and Suggestions for Authors
The present manuscript analyses the different content in catechin and rutin and the antioxidant activity of different apples. It would be desirable and more accurate to analyse the antioxidant activity also by ORAC and ABTS, and a deeper discussion of the results would be appreciated by the reader. In addition, in the text it is referred to certain supplementary material, but this referee has not been able to found it. Other points that should be checked:
We have already used DPPH and FRAP which are parallel assays for determining the anti-oxidant assays. But still we will work next year using ORAC and ABTS assays. We have revised some sections in results so as to provide lucid picture of our observations.
Please, provide the complete name before using an abbreviation.
We have already provided the complete name of the abbreviations like DPPH and FRAP, diphenyl-2-picrylhydrazyl (DPPH), and ferric reducing antioxidant power (FRAP) in Table 3 legend.
In table 1, what do the upper-index letters mean?
The upper index letters determine the level of significance in the different observations with respect to each other.
In certain occasion the authors refers to apple juice (also in the conclusion), but was not the extract of the fruit analysed? Please, explain.
We have determined all the parameters from the apple extract not apple juice as it is liable to fermentation.
Which was the purity of the standards (page 10, line 244)?
The standards procured from Sigma Aldrich having purity of 99.6%.

Reviewer 2 Report
In the present study, the amount of catechin and rutin in sixty varieties of apples were correlated to the antioxidant activity of the samples and significant variation in the antioxidant potential of different genotypes were highlighted.
The authors also suggest that combining correlation analysis and multivariate statistical techniques can represent a suitable approach to evaluate the quality of commercial fruits.
The rational of the study appears to be of interest; however, there are several points which require to be sorted and improved and which make the manuscript not suitable for publication in the present form.
One of the major limits is the phytochemical characterization of apple extracts. In fact, as stated in the introduction (lines 42-45), apple has been found to be a source of different phenolics and flavonoids, including chlorogenic, coumaric, coumaroylquinic, caffeic, ferulic, cinnamic, vanillic, gentisic, pro-tochatehuic, gallic acids, rutin, quercetin, catechin and procyanidins.
- How to make sure the functions are mainly due to rutin and catechin? Also, the possible contribution of other phytochemicals, phenolic acids and procyanidins should be evaluated.
- Is the extraction procedure suitable to concentrate these compounds? The HPLC metabolome of phenolics should be provided.
- In the introduction, the Authors stated that “the variation in catechin, rutin, quercetin content and antioxidant activity of apples was determined”, while only the amount of rutin and catechin was reported in Table 1.
- Rutin and quercetin ratio should be evaluated and discussed. Please, see and discuss “Di Sotto A. et al. Capsicum annuum L. var. Cornetto di Pontecorvo PDO: Polyphenolic profile and in vitro biological activities. Journal of Functional Foods, 40 (2018), 679-691”.
- Rewrite abstract and conclusion sections in order to highlights the novelty of this study.
Results section is poorly written, and data several mistaken data are reported.
Particularly,
- Line 75: the phenolic amount in Mallus floribunda ….by 980.8 GAE/g in Tydemans…please check the amount in the text and in the Table.
- Line 85: Table 1 should be replaced by Table 2, since the relative data are firstly discussed.
- Line 98: how many samples for each variety were analysed?
- Tables 1 and 2: data should be reported as mean ± standard deviation of multiple replicates.
- Lines 103-104: delete “2.3.1. Antioxidant Activity (DPPH) and Ferric Reducing Antioxidant Potential (FRAP) Assay for 104 free radical scavenging activity”
- Lines 105-107: rewrite
- Table 3: data analysed for Pearson correlation should be explained in order to clarify the meaning of the table. In the present form, it is not clear if data are representative of all varieties. This point should be explained also in the Methods.
- Line 143: the legend of Figure 1 should be improved in order to better describe the results.
- Line 219: how the apple genotypes were identified. Are there voucher specimens?
- Lines 244-245: delete “Quercetin, catechin and rutin standard were obtained from Sigma Aldrich.”
Author Response
Reviewer 2
Open Review
(x) I would not like to sign my review report
( ) I would like to sign my review report
English language and style
(x) Extensive editing of English language and style required
( ) Moderate English changes required
( ) English language and style are fine/minor spell check required
( ) I don't feel qualified to judge about the English language and style
Yes | Can be improved | Must be improved | Not applicable | |
Does the introduction provide sufficient background and include all relevant references? | ( ) | ( ) | (x) | ( ) |
Is the research design appropriate? | ( ) | (x) | ( ) | ( ) |
Are the methods adequately described? | ( ) | (x) | ( ) | ( ) |
Are the results clearly presented? | ( ) | ( ) | (x) | ( ) |
Are the conclusions supported by the results? | ( ) | (x) | ( ) | ( ) |
Comments and Suggestions for Authors
In the present study, the amount of catechin and rutin in sixty varieties of apples were correlated to the antioxidant activity of the samples and significant variation in the antioxidant potential of different genotypes was highlighted.
The authors also suggest that combining correlation analysis and multi-variate statistical techniques can represent a suitable approach to evaluate the quality of commercial fruits.
The rational of the study appears to be of interest; however, there are several points which require to be sorted and improved and which make the manuscript not suitable for publication in the present form.
One of the major limits is the phytochemical characterization of apple extracts. In fact, as stated in the introduction (lines 42-45), apple has been found to be a source of different phenolics and flavonoids, including chlorogenic, coumaric, coumaroylquinic, caffeic, ferulic, cinnamic, vanillic, gentisic, pro-tochatehuic, gallic acids, rutin, quercetin, catechin and procyanidins.
- How to make sure the functions are mainly due to rutin and catechin? Also, the possible contribution of other phytochemicals, phenolic acids and procyanidins should be evaluated.
Response: We have already demonstrated the DPPH and FRAP activity against two reference compounds like rutin and catechin in our ongoing projects. We have also determined phenolic and flavaniods in this project. We can provide the data for reference but it is unpublished data which is to be incorporated in our next manuscript.
- Is the extraction procedure suitable to concentrate these compounds? The HPLC metabolome of phenolics should be provided.
Response: We have optimized the HPLC process using different solvent for the procurement of the metabolites in the different genotypes.
- In the introduction, the Authors stated that “the variation in catechin, rutin, quercetin content and antioxidant activity of apples was determined”, while only the amount of rutin and catechin was reported in Table 1.
Response: It is typographical error we have not incorporated any data pertaining to quercetin. It will be omitted from the revised manuscript.
- Rutin and quercetin ratio should be evaluated and discussed. Please, see and discuss “Di Sotto A. et al. Capsicum annuum L. var. Cornetto di Pontecorvo PDO: Polyphenolic profile and in vitro biological activities. Journal of Functional Foods, 40 (2018), 679-691”.
Response: We have not determined quercetin in our current research. So discussing this topic rutin and quercetin ratio will be irrelevant in the light of the aforementioned paper will be irrelevant.
- Rewrite abstract and conclusion sections in order to highlights the novelty of this study.
Response: Abstract and conclusion is framed with clear and lucid observation so that it could provide inclusive picture of the paper subject.
Results section is poorly written, and data several mistaken data are reported.
Response: Rectified in revised manuscript.
- Line 75: the phenolic amount in Mallus floribunda ….by 980.8 GAE/g in Tydemans…please check the amount in the text and in the Table.
Response: It is typographical error and it has been revised and correct value is incorporated as the maximum content of phenolics 980.8 GAE/g was observed in wild apple genotype Mallus floribunda followed by 722.0 GAE/g in Tydemans Early Worcestor and minimum content of phenolics 31.5 mg L−1 was observed in Starking Delicious while as rest of genotypes are enlisted within moderate ranges.
- Line 85: Table 1 should be replaced by Table 2, since the relative data are firstly discussed.
Response: Table 1 cannot be replaced by Table 2 both are totally different. Table 1 deals with variability in rutin and catechin of different apple genotypes determined via HPLC while as Table 2 summarizes the variability in anti-oxidant efficacy of diverse apple genotypes.
- Line 98: how many samples for each variety were analysed?
Response: We have analyzed 10 samples (n=10) from each variety. All the samples were randomly selected.
- Tables 1 and 2: data should be reported as mean ± standard deviation of multiple replicates.
Response: We have included data in the format of mean ±standard deviation.
- Lines 103-104: delete “2.3.1. Antioxidant Activity (DPPH) and Ferric Reducing Antioxidant Potential (FRAP) Assay for 104 free radical scavenging activity”
Response: We have deleted 2.3.1 section in revised manuscript.
- Lines 105-107: rewrite
- Table 3: data analysed for Pearson correlation should be explained in order to clarify the meaning of the table. In the present form, it is not clear if data are representative of all varieties. This point should be explained also in the Methods.
Response: Correlation matrix is framed with respective to different assays such as DPPH and FRAP versus the total flavanoids, flavanols, the total phenolics, catechins and rutin so as to decipher the concrete relation between the metabolites and antioxidant potential determined via assays. We have not determined any relation with respect to apple varieties.
- Line 143: the legend of Figure 1 should be improved in order to better describe the results.
Response: The legend has been changed as suggested in revised manuscript.
- Line 219: how the apple genotypes were identified. Are there voucher specimens?
Response: The apple genotypes we have are registered with NBPGR and have specific IC/EC numbers. In case you need the IC/EC number of apple genotypes we can provide them as supplementary data.
- Lines 244-245: delete “Quercetin, catechin and rutin standard were obtained from Sigma Aldrich.”
Response: Changes are incorporated in the revised manuscript.

Reviewer 3 Report
The paper is interesting and regarding contribution to the field the manuscript is acceptable, methodology used is good, interpretation of results is acceptable.
I have one minor question
1.In the Discussion, the Authors should highlight the possible clinical significance of their findings.
Author Response
Reviewer 3
Open Review
(x) I would not like to sign my review report
( ) I would like to sign my review report
English language and style
( ) Extensive editing of English language and style required
(x) Moderate English changes required
( ) English language and style are fine/minor spell check required
( ) I don't feel qualified to judge about the English language and style
Comments and Suggestions for Authors
The paper is interesting and regarding contribution to the field the manuscript is acceptable, methodology used is good, interpretation of results is acceptable.
I have one minor question
1.In the Discussion, the Authors should highlight the possible clinical significance of their findings.
Response: Though we have not demonstrated any clinical studies on apple extracts but there are significant research studies which determine the medicinal value of apples. The clinical significance of catechin can be however correlated with anti-oxidant potential and radical scavenging activity of the apple samples. There is a significant disparity between the composition of phytochemicals like catechin and rutin in the different varieties of apple depending on the maturation and ripening stage of apple fruit. Different research experiments have determined the relationship between the incidences of varied diseases like cancer, coronary mortality with daily intake of apples. There is a significantly inverse relationship between flavoniod intake and lung cancer development. The relationship between dietary catechins and epithelial cancer has demonstrated 87% of the total catechin intake while apples contribute 8.0% of catechin consumption.

Round 2
Reviewer 1 Report
The revised manuscript has been improved, but there are some concerns that remain:
In table 1, please, state clearly the meaning of the upper-index (which letter indicates a significate difference with which group)
Please, note that the authors still refers to apple juice instead of extract fruit. Please, check.
Please, note the authors still refers to Figure S1, Table S1 and Table S2 instead of Figure 1 and so on. Please, check.
Please, note that the antioxidant activity many time does not correlate with a strong antioxidant effect in an organism, and that further studies are needed to confirm this aspect. Please, moderate the conclusion.
Author Response
Open Review
(x) I would not like to sign my review report
( ) I would like to sign my review report
English language and style
( ) Extensive editing of English language and style required
(x) Moderate English changes required
( ) English language and style are fine/minor spell check required
( ) I don't feel qualified to judge about the English language and style
Yes | Can be improved | Must be improved | Not applicable | |
Does the introduction provide sufficient background and include all relevant references? | (x) | ( ) | ( ) | ( ) |
Is the research design appropriate? | ( ) | (x) | ( ) | ( ) |
Are the methods adequately described? | (x) | ( ) | ( ) | ( ) |
Are the results clearly presented? | ( ) | (x) | ( ) | ( ) |
Are the conclusions supported by the results? | ( ) | (x) | ( ) | ( ) |
Comments and Suggestions for Authors
The revised manuscript has been improved, but there are some concerns that remain:
In table 1, please, state clearly the meaning of the upper-index (which letter indicates a significate difference with which group).
Response: The legends written on the table as superscript indicate significance with respect to each other (genotptes).
Please, note that the authors still refers to apple juice instead of extract fruit. Please, check.
Response: We have replaced the juice by the extract in the revised manuscript.
Please, note the authors still refers to Figure S1, Table S1 and Table S2 instead of Figure 1 and so on. Please, check.
Response: There is only table S1 and table S2 and figure S1 and S2 no longer exists now. Please see the revised manuscript.
Please, note that the antioxidant activity many time does not correlate with a strong antioxidant effect in an organism, and that further studies are needed to confirm this aspect. Please, moderate the conclusion.
Response: The changes have been incorporated in the revised manuscript.
The authors are grateful for all the comments.
Reviewer 2 Report
The manuscript has been improved and clarified by Authors, but some minor changes are still needed.
- Tables 1 and 2: include the number of replicates in parentheses (n =...)
- "Preparation of extracts" - line 235: insert "(10 fruits for each variety)" after "The fragmentation of apples"
- Conclusions - lines 296-297: please change the wording "catechins and rutin are the main contributors" as follow "catechins and rutin are involved in the antioxidant activity of the commercial juice under study, although the contribution of other phytochemicals cannot be excluded".
Author Response
Reviewer 2
(x) I would not like to sign my review report
( ) I would like to sign my review report
English language and style
( ) Extensive editing of English language and style required
( ) Moderate English changes required
( ) English language and style are fine/minor spell check required
(x) I don't feel qualified to judge about the English language and style
Yes | Can be improved | Must be improved | Not applicable | |
Does the introduction provide sufficient background and include all relevant references? | (x) | ( ) | ( ) | ( ) |
Is the research design appropriate? | (x) | ( ) | ( ) | ( ) |
Are the methods adequately described? | (x) | ( ) | ( ) | ( ) |
Are the results clearly presented? | (x) | ( ) | ( ) | ( ) |
Are the conclusions supported by the results? | (x) | ( ) | ( ) | ( ) |
Comments and Suggestions for Authors
The manuscript has been improved and clarified by Authors, but some minor changes are still needed.
- Tables 1 and 2: include the number of replicates in parentheses (n = 10) .
Response: The tables are represented in the form mean±SD with n=10.
- "Preparation of extracts" - line 235: insert "(10 fruits for each variety)" after "The fragmentation of apples"
Response: It has been incorporated in the revised manuscript.
- Conclusions - lines 296-297: please change the wording "catechins and rutin are the main contributors" as follow "catechins and rutin are involved in the antioxidant activity of the commercial juice under study, although the contribution of other phytochemicals cannot be excluded".
Response: The changes has been incorporated in the revised manuscript.
The authors are grateful for reviewers comments.